# Size Effect of Shear Strength of Recycled Concrete Beam without Web Reinforcement: Testing and Explicit Finite Element Simulation

**Wei Wang [1,2], Xin Zeng [1], Emery Niyonzima [1], Yue-Qing Gao [1,*], Qiu-Wei Yang [3] and Shao-Qing Chen [4]**

[1] College of Civil Engineering, Shaoxing University, Shaoxing 312000, China; wangwei1210@189.cn (W.W.); zx572761328@126.com (X.Z.); niyonzimaemery8@gmail.com (E.N.)

[2] Key Laboratory of Rock Mechanics and Geohazards of Zhejiang Province, Shaoxing University, Shaoxing 312000, China

[3] School of Civil and Transportation Engineering, Ningbo University of Technology, Ningbo 315211, China; yangqiuwei79@126.com

[4] State Grid Sichuan Electric Power Research Institute, Chengdu 610000, China; lizeshen@163.com

* Correspondence: gaoyueqing@usx.edu.cn

**Abstract:** Recycled concrete is a form of low-carbon concrete with great importance. The explicit finite element method is an economical and feasible method for analyzing static concrete structures, such as those made of recycled concrete. The shear strength of regular concrete beams has size effects. In this study, a group of physical tests on the size effect of the shear strength of recycled concrete beams without web reinforcement was carried out under the condition of a constant shear span ratio. The research results show that the shear strength of the test beam generally decreases with the increase in beam section height, and a regression formula of the shear strength was obtained, which can formulate this effect. The rationale and feasibility of the explicit finite element method solving the ultimate load of concrete structures (which can derive the shear strength) were briefly demonstrated, and an explicit finite element simulation of test beams was carried out. Results showed an obvious and phenomenologically regular size effect of the shear strength of recycled concrete beams without web reinforcement, which can be simulated by the explicit finite element method. This research aims to promote the study of low-carbon recycled concrete structures to a certain extent and encourage the application of economic explicit finite element methods for the static analysis of concrete structures.

**Keywords:** structural engineering; static analysis; test; explicit finite element method; low-carbon; recycled concrete beam; shear strength; size effect

## 1. Introduction

Recycled concrete is an important form of low-carbon-emission concrete; similarly, numerical simulation methods are generally regarded as low-carbon analysis methods compared with physical tests. The mass formed by waste concrete processing is commonly utilized as coarse aggregate, and the material formed by mixing it with inorganic binding substances, such as fine aggregate, cement, water, etc., is called recycled concrete (ReC) [1]. It is generally considered that ReC is a green, low-carbon-emission material, which has a certain positive role in the protection of resources and the environment [2]. It is worth noting that, from a broad perspective of processes in research, production and living, among multiple behaviors that can achieve the same goal, those producing lower greenhouse gas emissions are collectively called low-carbon behaviors [3]. Physical tests and numerical simulation tests are two common methods of studying building structures, and the finite element method is a regular means to carry out structural numerical simulation tests [4]. In most cases, the numerical simulation method is obviously a relatively low-carbon analysis method.

At present, recycled concrete material properties and the mechanical properties of components cast from ReC are research hotspots in civil engineering. Research on recycled concrete material properties mainly involves strength [5], carbonization performance [6], durability [7], and creep performance studies [8]. Studies on the mechanical properties of recycled concrete components mainly include the bending performance of the slab [9,10], the deformation performance [11], the compression performance [12] and the shearing performance [13] of the column, as well as the deformation performance [14], the bending performance [15] and the shearing performance [16–22] of the beam. In actual engineering, the main failure mode of concrete beams is shear failure. In order to make the beam meet the ultimate bearing capacity requirements, it is necessary to calculate the shear resistance of the beam [23]. Therefore, studies on the shear resistance of recycled concrete beams are of great significance.

From an engineering point of view, the shear resistance of concrete beams includes the itemized shear resistance of web reinforcement and the itemized shear resistance of concrete [24], and the latter is quantitatively affected by material characteristics. One of the inherent characteristics of recycled concrete, which is a quasi-brittle material, is the size effect [25], that is, the mechanical properties of a certain material depend on the specific geometric dimensions of the solid object made from it. In addition, it can be inferred that there is a size effect on the itemized shear resistance of concrete that makes up the overall shear resistance of recycled concrete beams. Hereinafter, the concrete itemized shear resistance of recycled concrete beams will be referred to as SRRC. A number of studies on the overall shear resistance of recycled concrete beams exist [16–21], as well as reports on the size effect of the concrete itemized shear resistance of original concrete beams [26–31]. However, research on the size effect of SRRC has been limited, with only one report by Zhao et al. found in the literature [22]. Although it is an experimental study on the size effect of SRRC under the condition of a constant shear span ratio, it provides only qualitative assessment and lacks the quantification of this effect. Besides, it does not use the finite element method for supplementary parameter analysis for limited experimental data. Therefore, it is necessary to carry out further physical tests on the size effect of the concrete itemized shear resistance (or SRRC) of recycled concrete beams.

The finite element method is a universal and economical numerical parameter analysis method for solving structural static problems in solid mechanics. If it is used reasonably and scientifically, it will assist with the study of the SRRC size effect. The essence of the finite element method is translating the solving of the problem of continuum mechanics described by a differential equation, into the solving of an approximately equivalent system of algebraic equations [32]. Compared with the physical test method, the finite element method can carry out structural analysis at a lower cost [33]. Obviously, this advantage is conducive to carrying out structural parameter analysis, which is an important method for reaching a comprehensive description of the characteristics of the research object [34]. R. Tartaglia et al. used the finite element parameter analysis method to study the internal force in the flange of the T-stub, the change law of the internal force in the bolt rod, the distance from the bolt hole to the flange edge, and flange bending and other manufacturing defects that influence the mechanical behavior of the profile [35].

In finite element static analysis of concrete structures, the system of algebraic equations to be solved is nonlinear. At present, implicit and explicit algorithms are commonly used to solve the incremental form of this system [36]. Since the damage constitutive model can better describe the mechanical behavior of concrete materials, such as the mechanical phenomenon of strain-softening when concrete is cracked or crushed [37], it is often used when constructing concrete structure models [38]. In the above case, the algebraic systems describing the equilibrium are nonlinear. The static incremental step strategy is commonly used to approximate the equilibrium path [32]. At present, the algorithms for solving incremental steps mainly include implicit algorithms (the so-called Newton-like algorithms) and explicit algorithms [39]. The essence of an implicit algorithm is to use a kind of iterative method to directly solve the algebraic system of static balance described

in the incremental form [40]. The implicit algorithm is often recommended as a general method to solve nonlinear problems in commercial finite element software [41]. One of its limitations is that, when solving structural static problems with local instability phenomena (such as cracking when concrete structures are forced), it is generally difficult to obtain a convergent solution [42]. For concrete beams, when the external load has approached the peak load of SRRC [23], inclined cracks have already appeared in the shear area of the concrete beam. From the foregoing of two points, it is easy to infer that it is difficult for the implicit algorithm to research the SRRC size effect. The essence of the explicit algorithm is to convert the static problem of the original structure into the corresponding structural dynamic problem, perform a pseudo-static analysis on this problem, and approximate the result of the pseudo-static analysis as the solution to the original static problem [43]. The static problem that the explicit algorithm can approximately solve does not depend on the material, geometry, and continuity characteristics of the problem at all [44]. To date, there have been many reports on the use of explicit algorithms to carry out approximate static analysis of concrete structures [45–47]. For example, Yao et al. [46] used an explicit finite element method to analyze the four-point bending beam of concrete in order to verify the reliability of the explicit algorithm in solving the quasi-static response of concrete beams. There is good agreement with the test results. Yu et al. [47] used the arch effect to explain the shear mechanism of variable cross-section beams, conducted finite element analysis on four cantilever beams without web reinforcement, proposed the influence coefficient of the compression inclination on the arch effect and the method to determine the position of the check section, and established the relationship between the standard formula for shear resistance and the formula for calculating the shear resistance of beams with variable cross-sections. However, no study exists on the use of explicit algorithms to investigate the size effect of SRRC. Therefore, the present study aims to use the explicit finite element method to carry out simulation experiments on the SRRC size effect.

In summary, the main contribution of this work is the testing and explicit finite element simulation of the shear strength of a group of recycled concrete beams without web reinforcement under the condition of a constant shear span ratio.

The purpose of this paper is to report that SRRC (the concrete itemized shear resistance of recycled concrete beams) has a size effect, and this effect can be simulated by the explicit finite element method but is difficult to simulate using the implicit finite element method.

The paper is structured as follows. An overview of the physical tests is provided in Section 2, including the specimen design, the loading equipment, and the loading system. In Section 3, an overview of the simulation tests is given, including the physical discrete and contact settings, material constitutive model selection, loading system, and solution algorithm. In Section 4, some typical test results and related discussions are provided first. Next, a regression formula is presented, which can reflect the size effect of shear strength. Section 5 concludes this paper.

## 2. Overview of Physical Tests

### 2.1. Specimen Design

A total of four recycled concrete beams numbered B120, B180, B240, and B300 were designed and poured with the same batch of recycled concrete. Before pouring the 4 test specimens, the test materials and test mix ratios need to be determined. They are introduced as follows [22].

Firstly, the test materials include natural fine aggregate, recycled coarse aggregate and cement.

In this study, the performance indicators for natural fine aggregate are presented in Table 1.

**Table 1.** Parameters of fine aggregate.

| Fine Aggregate | Particle Size/mm | Apparent Density/(kg/m³) | Bulk Density/(kg/m³) | Moisture Content/% | Fineness Modulus Number/μf | Water Absorption/% |
|---|---|---|---|---|---|---|
| Yellow sand | <5 | 2548 | 1211 | 6.8 | 1.83 | 2.9 |

When determining the recycled coarse aggregate, certain factors are also mainly taken into account, such as density, water absorption, moisture and size. For details of the performance indicators of the coarse recycled aggregate selected in this study, see Table 2.

**Table 2.** Parameters of recycled coarse aggregate.

| Apparent Density/(kg/m³) | Bulk Dense/(kg/m³) | Water Absorption/% | Moisture Content/% | Crush Value Index/% | Porosity/% | Porosity/% | Pudding Rate/% |
|---|---|---|---|---|---|---|---|
| 2481 | 1240 | 6.3 | 2.2 | 19.9 | 5.7 | 51.0 | 42.9 |

When determining the cement, some factors are mainly considered compressive, such as density. The cement used in this test is ordinary Portland cement with a strength level of 32.5. For some performance indicators of the cement selected in this study, see Table 3.

**Table 3.** Parameters of cement.

| Density/(kg/m³) | Compressive Strength/MPa | | Flexural Strength/MPa | |
|---|---|---|---|---|
| | 3d | 28d | 3d | 28d |
| 3051 | 17.8 | 35 | 3.9 | 7.1 |

Secondly, the mix ratio is designed. The ratio used to produce ReC, the strength level of which is C30, the slump is 70 mm, and which was used to cast the 4 beams tested in this study, is given by Table 4.

**Table 4.** Recycled concrete mix design.

| Replacement Rate/% | Water-Cement Ratio | Material Consumption Per Cubic Meter/(kg/m³) | | | |
|---|---|---|---|---|---|
| | | Cement | Water | Sand | Recycled Coarse Aggregate |
| 100 | 1:0.34 | 623.3 | 211.8 | 491.4 | 1073.5 |

After testing, the average uniaxial compressive strength of this recycled concrete was

$$\sigma_{cp} = 19.88 \text{ MPa}, \tag{1}$$

and the initial elastic modulus was

$$E_c = 2.588 \times 10^4 \text{ MPa}. \tag{2}$$

Based on the purpose of the test, the test beam was not equipped with stirrups. However, so that insufficient bending strength would not affect the results of the shear strength test, two longitudinal bars were arranged at the bottom of the beam. The position, yield strength ($f_y$), elastic modulus ($E_s$) and area ($A_s$) information of the longitudinal bars in each beam, and the size information of each beam, are shown in Figure 1 and Table 5. Only two supporting points of the test beam were located at the bottom of the beam, and the form was simply supported; the only concentrated force loading point was at the top of the beam. The specific positions of the supporting point and the loading point are shown in Figure 1 and Table 5. It is easily deduced that the beam's shear span ratio is $\lambda = l_2/h_0 = 2.45$, where $l_2$ is the horizontal distance between the supporting point and

the loading point, and $h_0$ is the effective height. After the estimation and according to the above-mentioned geometric dimensions, steel bar configuration, material strength and loading design, the failure mode of recycled concrete beams will be guaranteed to be shear failure, and thus the research purpose can be achieved.

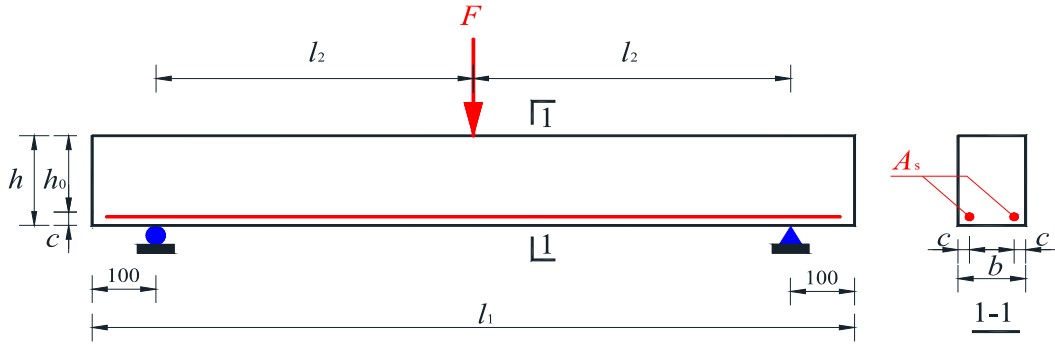

**Figure 1.** Geometrical properties of specimens.

**Table 5.** The dimensions of specimens and the properties of steel bars.

| Specimen | $l_1$/mm | $l_2$/mm | $b$/mm | $h$/mm | $h_0$/mm | $A_s$/(mm$^2$) | $f_y$/MPa | $E_s$/MPa |
|---|---|---|---|---|---|---|---|---|
| B-120 | 740 | 270 | 120 | 120 | 110 | 157.00 | 481 | $1.96 \times 10^5$ |
| B-180 | 1010 | 405 | 120 | 180 | 165 | 307.72 | 459 | $1.89 \times 10^5$ |
| B-240 | 1280 | 540 | 120 | 240 | 220 | 401.92 | 421 | $2.12 \times 10^5$ |
| B-300 | 1550 | 675 | 120 | 300 | 275 | 508.68 | 414 | $2.12 \times 10^5$ |

### 2.2. Loading Equipment and Loading System

The loading details of the test are depicted in Figure 2, where Figure 2a shows the reaction frame, and Figure 2b includes the details of the loading configuration. During loading, the preset loading schedule was implemented by simultaneously 'controlling the oil output rate of the oil pump to slowly move the jack lifting sleeve down, and monitoring the force value of the force indicator connected to the force sensor'.

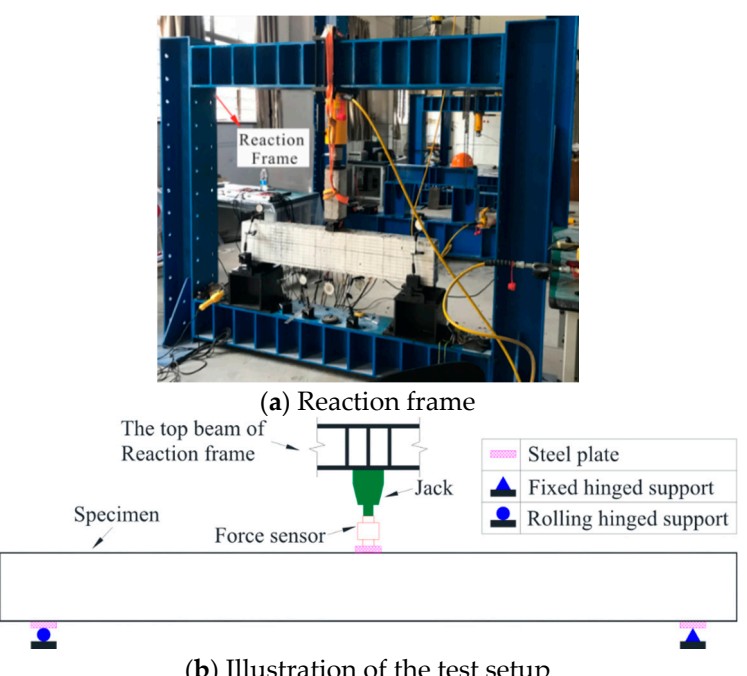

(**a**) Reaction frame

(**b**) Illustration of the test setup

**Figure 2.** Test setup.

The whole loading history was divided into two stages (pre-loading and formal loading) with different loading schedules. Before formal loading, preload was applied at about 50% of the estimated cracking load, $F_{cr,c} = M_{cr,c}/l_2$, where the cracking moment for the flexural load capacity of a normal section of beams without web reinforcement, $M_{cr,c}$, was calculated according to the formula given in the literature [24]:

$$M_{cr,c} = 0.292[1 + 5 \times A_s E_s/(bhE_c)]\sigma_{tp}bh^2.$$

In the formula, $\sigma_{tp}$ is the peak tensile stress, and based on the formula derived from literature [24] and literature [48],

$$\sigma_{tp} = 0.395 \times (\sigma_{cp}/0.715)^{0.55}. \tag{3}$$

After calculating, $\sigma_{tp}$ equals to 2.46 MPa. The formal loading phase was divided into three subphases. Firstly, the increment of $\Delta F_1 \approx 0.1F_{cr,c}$ was used to increase the concentrated load $F$ from 0 to the actual cracking load, $F_{cr,c}$. Secondly, $\Delta F_2 \approx 0.1(F_{q,c} - F_{cr,t})$ was applied to increase $F = F_{cr,t}$ approximately to $F_{q,c}$. The latter, $F_{q,c}$, is the quantile value of the estimated ultimate load ($F_{u,c}$) and in this study is equal to $0.7F_{u,c}$. This paper used the formula $V_{u,c} = 1.75\sigma_{tp}bh_0/(\lambda + 1)$ to estimate $F_{u,c} = 2V_{u,c}$, given by the literature [25] for the shear bearing capacity of the oblique section of beams without web reinforcement. Thirdly, $\Delta F_3 \approx 0.1(F_{u,c} - F_{q,c})$ was used to increase $F = F_{q,c}$ to the measured limit load, $F_{u,t}$. $M_{cr,c}$, $F_{cr,c}$, $F_{u,c}$, $F_{q,c}$, $\Delta F_1$, $\Delta F_2$, and $\Delta F_3$ were calculated according to the above principles and $F_{cr,t}$, as seen in Table 6, thus determining the load control points during the entire test loading history.

**Table 6.** Several important estimated loads and internal forces.

| Specimen ID | $M_{cr,c}$/(kN·m) | $F_{cr,c}$/kN | $\Delta F_1$/kN | $F_{cr,t}$/kN | $F_{u,c}$/kN | $F_{q,c}$/kN | $\Delta F_2$/kN | $\Delta F_3$/kN |
|---|---|---|---|---|---|---|---|---|
| B-120 | 1.78 | 6.60 | 1.00 | 16.00 | 33.43 | 23.40 | 1.00 | 1.00 |
| B-180 | 4.31 | 10.70 | 1.00 | 18.00 | 50.15 | 35.11 | 1.00 | 1.00 |
| B-240 | 7.93 | 14.70 | 1.00 | 30.00 | 66.87 | 46.81 | 1.00 | 2.00 |
| B-300 | 12.40 | 18.40 | 2.00 | 26.30 | 83.59 | 58.51 | 3.00 | 2.00 |

## 3. Overview of Simulation Test

### 3.1. Physical Discrete Setting and Contact Conditions

In this paper, Abaqus software was used to carry out the geometrical discrete and contact settings of the finite element model of the specimen. The operation mainly involved six aspects. Firstly, the element type was determined. Both the beam entities and the steel cushion blocks were discrete, with an eight-node reduced integral format three-dimensional entity element. The number for this element in the Abaqus element library is C3D8R. Reinforcing bars were discretized by three-dimensional two-node line elements, with the number T3D2. Secondly, the element size was determined. For the rectangular solid elements of discrete steel cushion blocks and beam entities, the three orthogonal edges on the surface were parallel to the three orthogonal edges on the beam surface (see Figure 3), and the dimension of $x_l$ along the beam length was $l_e$, the dimension of the direction $x_h$ along the beam height was $h_e$, and the dimension of $x_b$ along the beam width was $b_e$. For the concrete element in the column of the cushion blocks, however, the dimension in the $x_l$ direction was $l_{e,in}$; for the concrete element from the beam end to the supporting cushion block area, the dimension in the $x_l$ direction was $l_{e,out}$. The dimensions of all the above physical elements are shown in Table 7. For the linear element of the discrete steel bars, the dimension $l_{es}$ along the beam length direction $x_l$ is also shown in Table 7, and its position in the plane $x_b$-$x_l$ is shown in Figure 1. Thirdly, it was determined how steel and concrete interact. This article assumed that there was no bond-slip behavior between the steel bar and the concrete, and thus the 'Embedded' constraint was selected to embed the steel bar in the concrete entity. Fourthly, the contact mode between the steel

cushion block and the concrete entity was determined. Herein, the mechanical behaviors of the upper and lower neighborhoods of the contact surface between the cushion blocks and the beam are set as follows: the contact surface between the left and right supporting cushion block and the bottom of the beam, and the contact surface between the loading cushion block and the top of the beam, adopted the 'Tie' mode to combine into a continuous medium area. Fifthly, boundary conditions were identified. In order to better simulate actual working conditions, the lower surface of the left and right supporting cushion block only had rotational degrees of freedom around $x_b$ and $x_h$. Sixthly, the constitutive response law of the neighboring concrete at the concentrated force was assumed. Extremely high stresses occur in concrete elements near the supporting/loading cushion blocks. In order to prevent these from exceeding the description range of the concrete damage constitutive model and subsequently causing calculation failures, the constitutive model of the steel cushion block was adopted for the concrete elements that are in contact with the left and right supporting cushion blocks and for the concrete element in contact with the loading cushion block (as shown in the red area in Figure 3). The mathematical expression of this model is shown in Section 3.2. According to the above six operational steps, a three-dimensional view of the finite element model of the beam, numbered B-120, is shown in Figure 3.

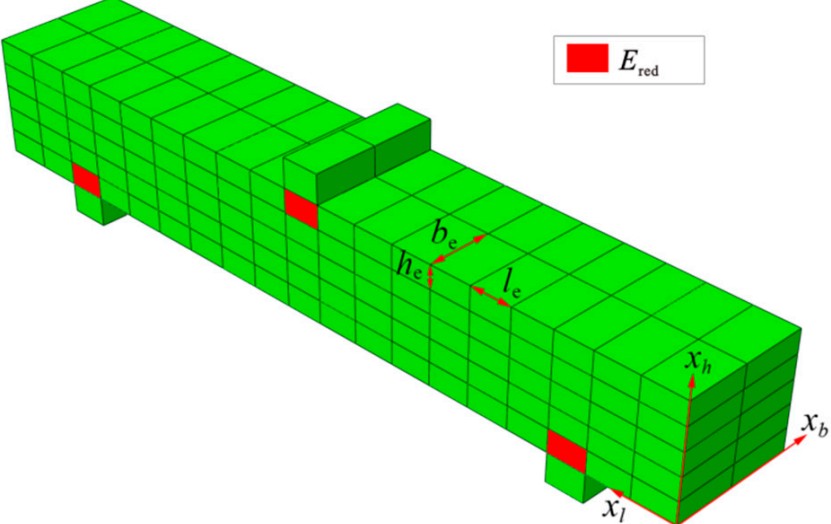

**Figure 3.** Finite element model of tested beam B-120. (Note: $E_{red}$ is the concrete elements that used the constitutive model of the steel cushion block).

**Table 7.** Dimensions of each element in finite element models of specimens/mm.

| Specimen ID | Concrete Element | | | | | Steel-Pad Element | | | Steel-Bar Element |
|---|---|---|---|---|---|---|---|---|---|
| | $l_e$/mm | $l_{e,in}$/mm | $l_{e,out}$/mm | $h_e$/mm | $b_e$/mm | $l_e$/mm | $h_e$/mm | $b_e$/mm | $l_{es}$/mm |
| B-120 | 39 | 36 | 41 | 24 | 60 | 36 | 30 | 60 | 40 |
| B-180 | 50.6 | 50.5 | 74.8 | 30 | 60 | 50.5 | 30 | 60 | 51.6 |
| B-240 | 48.9 | 51.2 | 74.4 | 40 | 60 | 51.2 | 30 | 60 | 49.6 |
| B-300 | 61.3 | 60.2 | 69 | 50 | 60 | 62.0 | 30 | 60 | 62.5 |

### 3.2. Material Constitutive Model

Before static analysis, constitutive models of various components in the finite element discrete model need to be selected.

Firstly, the constitutive model of the steel cushion blocks was selected. According to the local influence principle [49], the selection of the constitutive model of the steel cushion block has a limited effect on the simulation results of the static behavior of the concrete

beam. Therefore, in static loading, it can be assumed that the steel pad is always in a linear elastic state. Therefore, this study selected a linear elastic constitutive model for the steel cushion block, which can be expressed as

$$\sigma = \boldsymbol{D}^{\mathrm{e}} : \varepsilon,$$

where $\sigma$ is the Cauchy stress, $\varepsilon$ is the Cauchy strain tensor, and $\boldsymbol{D}^{\mathrm{e}}$ is the elastic stiffness tensor of the material, which is determined by the elastic modulus $E_b$ and the Poisson's ratio $\nu_b$ of the steel cushion block [49]. In the simulation of this study, $E_b = 190\,\mathrm{GPa}$, and $\nu_b = 0.3$.

Secondly, the constitutive model of steel bars was selected. The simulated beam was a simply supported beam without web reinforcement. When it failed due to the insufficient shear strength of the oblique section, the longitudinal reinforcement had not yet generally entered the yield stage. Therefore, this study employed an ideal (one-dimensional) elastoplastic constitutive model for steel bars, which can be expressed as

$$\begin{cases} \sigma_{\mathrm{s}} = E_{\mathrm{s}}\varepsilon_{\mathrm{s}} & \varepsilon_{\mathrm{s}} < \varepsilon_{\mathrm{sy}} \\ \sigma_{\mathrm{s}} = f_{\mathrm{sy}} & \varepsilon_{\mathrm{s}} \geq \varepsilon_{\mathrm{sy}} \end{cases},$$

where $\sigma_{\mathrm{s}}$ and $\varepsilon_{\mathrm{s}}$ indicate stress scalar and strain scalar, respectively; and $\varepsilon_{\mathrm{sy}} = f_y / E_{\mathrm{s}}$ is the initial yield strain, where $f_y$ stands for the yield strength. In the simulation, the $E_{\mathrm{s}}$ and $f_y$ of the longitudinal reinforcement in each simulated beam adopt the actual measured values of the longitudinal reinforcement in the corresponding test beam; see Table 5 for details.

Thirdly, the recycled concrete constitutive model was chosen. ① In the static test, there is a cracking phenomenon in the concrete, strain softening in the cracked area, and elastic-plastic strain unloading in the adjacent cracked area. Accordingly, this paper used the concrete damage plasticity (hereinafter referred to as CDP) constitutive model proposed by J. Lee et al. [50] for recycled concrete, which can describe the above behaviors and can be expressed as

$$\sigma = (1 - d)\boldsymbol{D}^{\mathrm{e}} : (\varepsilon - \varepsilon^{\mathrm{P}}), \tag{4}$$

where $d$ is the damage variable (scalar) and $\varepsilon^{\mathrm{P}}$ is the plastic strain tensor. According to this constitutive model theory [51,52], in the analysis, the stress-strain curve under the uniaxial (tension or compression) condition, the equivalent compression plastic strain $\tilde{\varepsilon}_{\mathrm{c}}^{\mathrm{p}}$, and the equivalent tension plastic strain $\tilde{\varepsilon}_{\mathrm{t}}^{\mathrm{p}}$ should be given. With the aid of several conversion rules, Formula (4) can be utilized to calculate the constitutive state of the integration point in the element. ② The literature [53] has pointed out that the stress-strain curve shapes of recycled concrete and original concrete are highly similar. Therefore, this study applied the uniaxial compressive stress-strain curve of concrete, as proposed by Guo [54],

$$\begin{cases} \sigma_{\mathrm{c}} = \sigma_{\mathrm{cp}}\left[\alpha_{\mathrm{c}}\frac{\varepsilon_{\mathrm{c}}}{\varepsilon_{\mathrm{cp}}} + (3 - 2\alpha_{\mathrm{c}})\left(\frac{\varepsilon_{\mathrm{c}}}{\varepsilon_{\mathrm{cp}}}\right)^2 + (\alpha_{\mathrm{c}} - 2)\left(\frac{\varepsilon_{\mathrm{c}}}{\varepsilon_{\mathrm{cp}}}\right)^3\right] & (\varepsilon_{\mathrm{c}} \leq \varepsilon_{\mathrm{cp}}) \\ \sigma_{\mathrm{c}} = \sigma_{\mathrm{cp}}\left(\frac{\varepsilon_{\mathrm{c}}}{\varepsilon_{\mathrm{cp}}}\right) / \left[\beta_{\mathrm{c}}\left(\frac{\varepsilon_{\mathrm{c}}}{\varepsilon_{\mathrm{cp}}} - 1\right)^2 + \frac{\varepsilon_{\mathrm{c}}}{\varepsilon_{\mathrm{cp}}}\right] & (\varepsilon_{\mathrm{c}} > \varepsilon_{\mathrm{cp}}) \end{cases}, \tag{5}$$

and the uniaxial tensile stress-strain curve

$$\begin{cases} \sigma_{\mathrm{t}} = \sigma_{\mathrm{tp}}\left[\alpha_{\mathrm{t}}\frac{\varepsilon_{\mathrm{t}}}{\varepsilon_{\mathrm{tp}}} + \left(\frac{6 - 5\alpha_{\mathrm{t}}}{4}\right)\left(\frac{\varepsilon_{\mathrm{t}}}{\varepsilon_{\mathrm{tp}}}\right)^2 + \left(\frac{\alpha_{\mathrm{t}} - 2}{4}\right)\left(\frac{\varepsilon_{\mathrm{t}}}{\varepsilon_{\mathrm{tp}}}\right)^6\right] & (\varepsilon_{\mathrm{t}} \leq \varepsilon_{\mathrm{tp}}) \\ \sigma_{\mathrm{t}} = \sigma_{\mathrm{tp}}\left(\frac{\varepsilon_{\mathrm{t}}}{\varepsilon_{\mathrm{tp}}}\right) / \left[\beta_{\mathrm{t}}\left(\frac{\varepsilon_{\mathrm{t}}}{\varepsilon_{\mathrm{tp}}} - 1\right)^{\varphi_{\mathrm{t}}} + \frac{\varepsilon_{\mathrm{t}}}{\varepsilon_{\mathrm{tp}}}\right] & (\varepsilon_{\mathrm{t}} > \varepsilon_{\mathrm{tp}}) \end{cases} \tag{6}$$

to construct the basic data required for numerical implementation of the above damage constitutive model. In the above two formulas, $\sigma_{\mathrm{c}}$ and $\varepsilon_{\mathrm{c}}$ are the compressive stress and compressive strain, respectively; $\sigma_{\mathrm{cp}}$ is the peak compressive stress; $\varepsilon_{\mathrm{cp}}$ is the strain corresponding to $\sigma_{\mathrm{cp}}$; $\sigma_{\mathrm{t}}$ and $\varepsilon_{\mathrm{t}}$ are the tensile stress and tensile strain, respectively; $\sigma_{\mathrm{tp}}$ is the peak tensile stress; $\varepsilon_{\mathrm{tp}}$ is the strain corresponding to $\sigma_{\mathrm{tp}}$; and $\alpha_{\mathrm{c}}$, $\beta_{\mathrm{c}}$, $\alpha_{\mathrm{t}}$, $\beta_{\mathrm{t}}$ and $\varphi_{\mathrm{t}}$

are all coefficients calibrated by the tests. Based on the tested $\sigma_{cp}$ (the value is shown in Formula (1)), first calculate $\sigma_{tp}$ according to Formula (2), and then calculate $\varepsilon_{tp}$ according to the formula $\varepsilon_{tp} = 65\sigma_{tp}^{0.54} \times 10^{-6}$ [54]. The values of these three data are listed in Table 8. At the same time, according to [53,54], the selected or calculated values of $\varepsilon_{cp}$, $\alpha_c$, $\beta_c$, $\alpha_t$, $\beta_t$ and $\varphi_t$ are also listed in Table 8. According to Formulas (5) and (6) and the parameters in Table 8, the stress-strain curve is presented in Figure 4. ③ As suggested by the literature [52], the formula

$$\begin{cases} \widetilde{\varepsilon}_c^p = \gamma_c(\varepsilon_c - \sigma_c/E_c) \\ \widetilde{\varepsilon}_t^p = \gamma_t(\varepsilon_t - \sigma_t/E_c) \end{cases}$$

is chosen to describe $\widetilde{\varepsilon}_c^p$ and $\widetilde{\varepsilon}_t^p$. In the formula, $E_c$ is the initial elastic modulus of concrete (its value is shown in Formula (2)), and $\gamma_c$ and $\gamma_t$ are the coefficients calibrated by the experiment, empirically taking the values of $\gamma_c = 0.7$ and $\gamma_t = 0.1$, based on the literature [52].

**Table 8.** Parameters for the stress-strain curves in the concrete constitutive model.

| $\sigma_{cp}$/MPa | $\varepsilon_{cp}$ | $\sigma_{tp}$/MPa | $\varepsilon_{tp}$ | $\alpha_c$ | $\beta_c$ | $\alpha_t$ | $\beta_t$ | $\varphi_t$ |
|---|---|---|---|---|---|---|---|---|
| 19.88 | $1.80 \times 10^{-3}$ | 2.46 | $1.06 \times 10^{-4}$ | 2.34 | 4.0 | 1.1 | 1.89 | 1.7 |

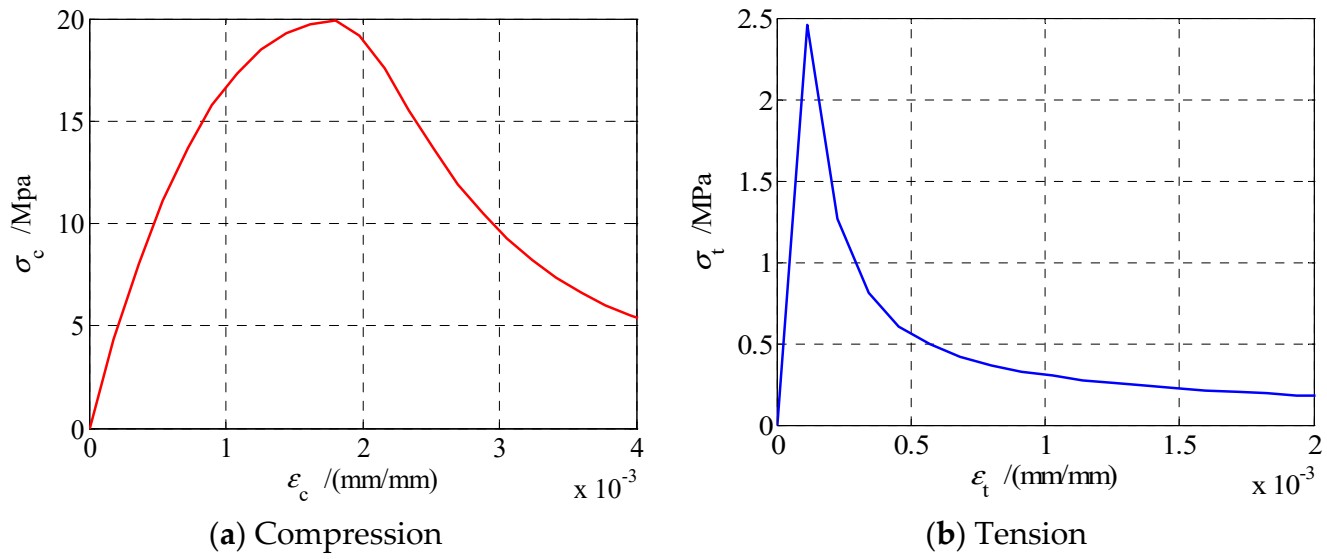

(**a**) Compression      (**b**) Tension

**Figure 4.** $\sigma$-$\varepsilon$ curves.

In order to increase the readability of the paper, follow the data input requirements of the CDP model in Abaqus. Four curves on the relationships of yield-stress (under compression) to inelastic-strain, compression-damage to inelastic-strain, yield-stress (under tension) to cracking-strain, and tension-damage to cracking-strain are shown in Figure 5.

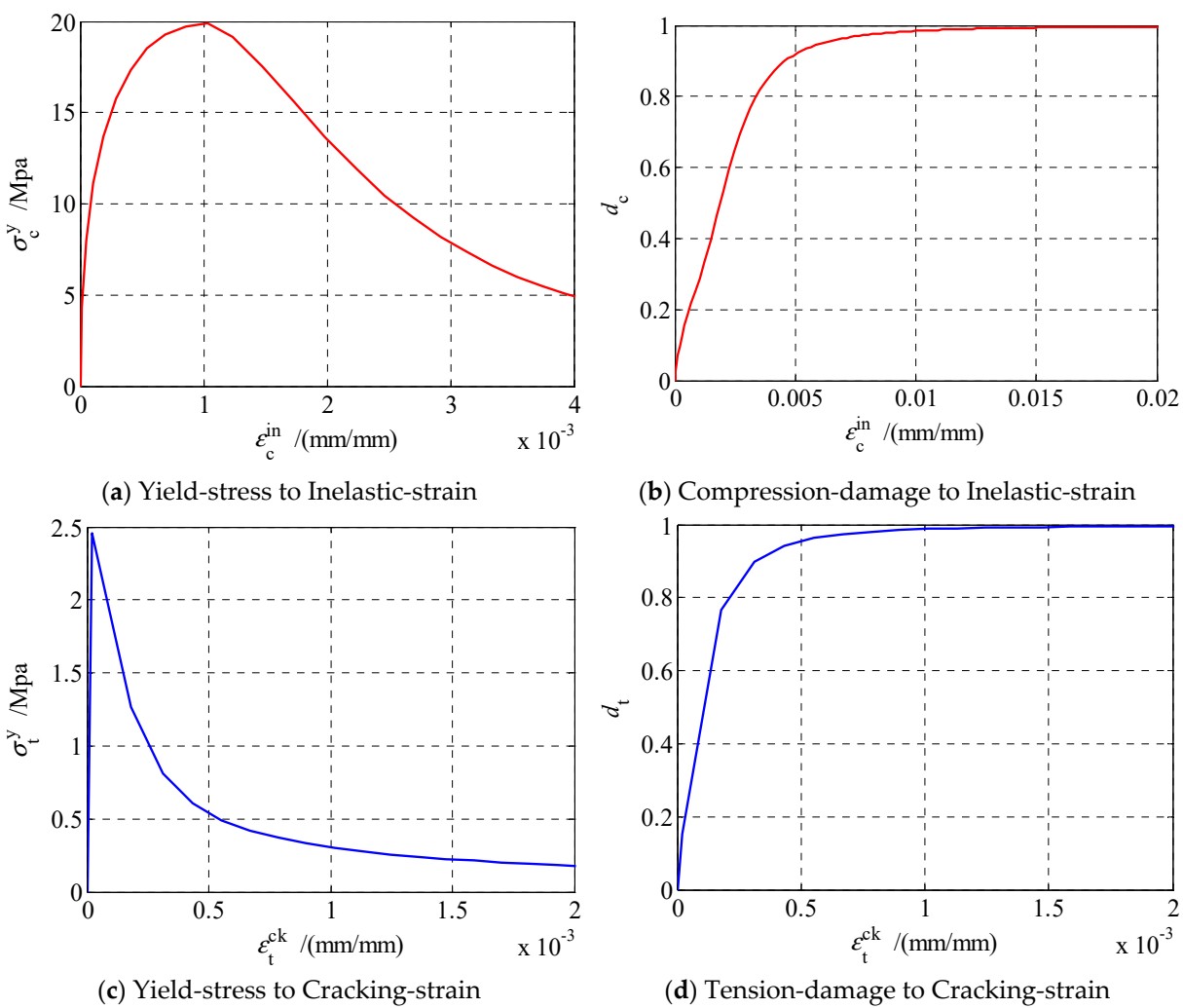

(**a**) Yield-stress to Inelastic-strain
(**b**) Compression-damage to Inelastic-strain
(**c**) Yield-stress to Cracking-strain
(**d**) Tension-damage to Cracking-strain

**Figure 5.** Data input requirements of the concrete damage plasticity (CDP) model in Abaqus. (Note: in the Figure, $\sigma_c^y$ is yield-stress under compression, $\varepsilon_c^{in}$ is inelastic-strain, $d_c$ is compression-damage, $\sigma_t^y$ is yield-stress under tension, $\varepsilon_c^{in}$ is cracking-strain, and $d_t$ is tension-damage).

### 3.3. Loading System and Solution Algorithm Configuration

Before performing the finite element simulation, it is also necessary to define the loading schedule and solving algorithm. In a similar way to the physical test, the displacement control loading mode is adopted in the simulation test, with a constant loading rate. The total loading time of each numerical beam model and the initial displacement/final displacement of the active loading point are shown in Table 9.

**Table 9.** Loading configuration.

| Specimen ID | Total Loading time/s | Initial Loading Displacement/mm | Final Loading Displacement/mm | Displacement Changing Law |
|---|---|---|---|---|
| B-120 | 2 | 0.0 | 0.20 | Linear |
| B-180 | 2 | 0.0 | 0.20 | Linear |
| B-240 | 2 | 0.0 | 0.40 | Linear |
| B-300 | 2 | 0.0 | 0.50 | Linear |

Configure the parameters of the implicit/explicit solver. For the implicit solver, the time period was set to 1, the loading point displacement changed linearly with time, the "Nlgeom" option was checked, the initial increment size was set to 0.001, the minimum

increment size was set to $1 \times 10^{-5}$, the maximum increment size was set to 0.01, and the iterative solution of the nonlinear equation system was Newton's method. For the explicit solver, the time period and the characteristics of loading point displacement in the time domain are shown in Table 9; check the "Automatic" option to determine the loading point increment, check the "Element-by-element" option to determine the stable increment step size, the time scaling factor was set to 1, the linear bulk viscosity parameter was set to 0.06, the quadratic bulk viscosity parameter was set to 1.2, and other parameters used the default values.

## 4. Results and Discussion

### 4.1. Failure Phenomenon and Characteristic Stress Contour Plots

The progress of crack development in the four beams was consistent with common sense, and the damage mode was typically uniform. When the load reached a certain level, normal transverse bending cracks first appeared in the area of maximum bending moment (mid-span). As the load increased, the area with normal bending cracks appearing enlarged, and the already existing normal bending cracks elongated. When the load continued to rise, shear oblique cracks appeared near the neutral axis of the beam, the oblique cracks extended to both ends, and part of the oblique cracks connected with the normal cracks. As the load reached an extreme value, a main diagonal crack suddenly appeared to tear the beam in a similar manner to the main control crack (the red curve) in beam B-240, as shown in Figure 6a, and the bearing capacity of the beam was lost. The progress diagram of the cracks (the curve in the figure) in beams B-120, B-180, and B-300 are shown in Figure 6b–d, respectively, in which the red curve refers to the main control crack. To determine the plane position of the cracks, an equidistant grid was drawn along the horizontal and vertical directions of the beam, and the grid number was marked with numbers and letters. The value near several positions of the crack represents the value of the applied load (unit: kN) corresponding to the crack developed to reach this point. The crack progress diagram of beam B-240 was consistent with the characteristics of the first three beams.

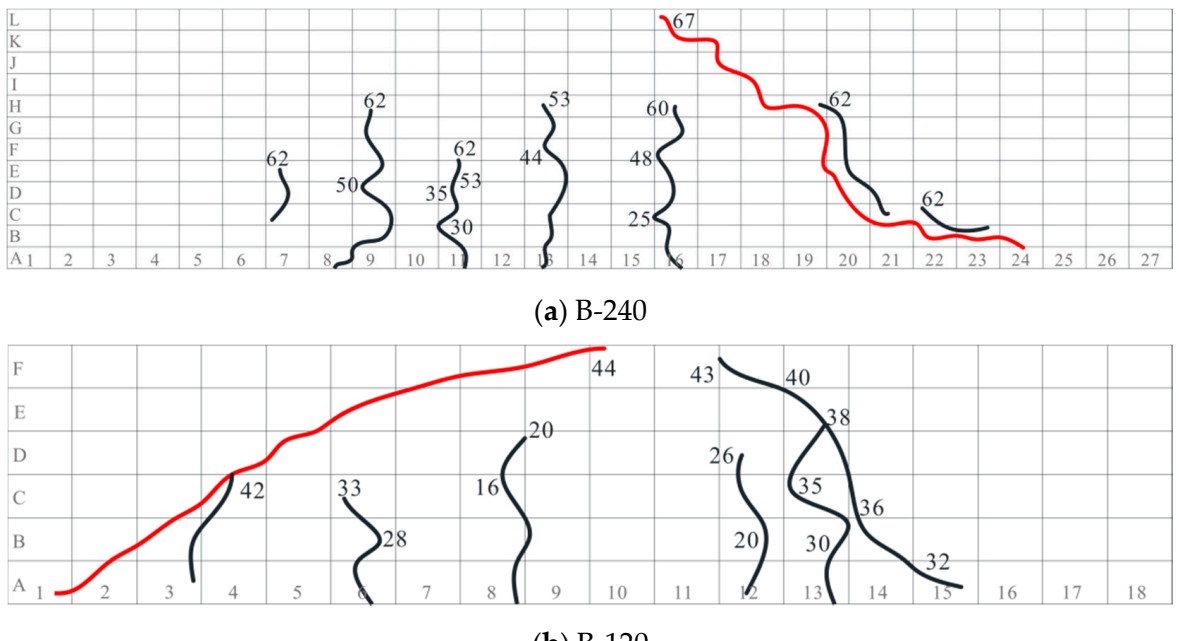

(**a**) B-240

(**b**) B-120

**Figure 6.** *Cont.*

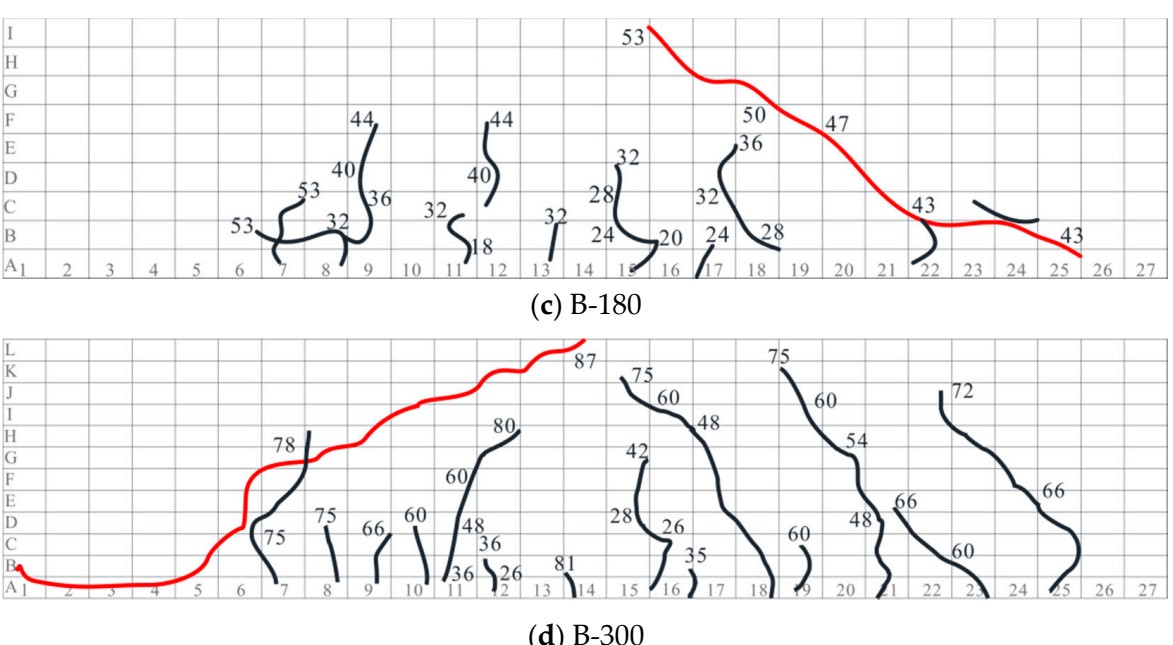

**(c)** B-180

**(d)** B-300

**Figure 6.** The scenes of specimen cracking.

The typical stress contour plots provided by the finite element simulation can verify the aforementioned experimental phenomenon. The law of the main tension stress contour plots given by the finite element simulation was similar for all four beams, while this paper only selects the beam B-240 for analysis. In basic terms, Figure 7a shows the main tensile stress contour plot of the beam after removing the cushion blocks and the element immediately adjacent to the cushion blocks. From this figure, it is clear that there is a high principal tensile stress in the high-stress area near the cushion blocks and the high-bending effect area at the bottom of the mid-span beam, which is qualitatively consistent with the common sense of elasticity and material mechanics; the principal tensile stress in the vicinity of the neutral axis of the beam is substantial. This phenomenon can also be qualitatively described by the theory of material mechanics—the shear stress here is large, but the normal stress is zero. Subsequently, the distribution of the aforementioned shear stress peak area can also be described by the contour map of Tresca stress $\tau_{max}$, as shown in Figure 7b ($\tau_{max} = (\sigma_1 - \sigma_3)/2$), where $\sigma_1$ and $\sigma_3$ are the maximum and minimum principal stresses, respectively. In particular, in the bending neutral axis area with high shear stress but low normal stress, the area with a large $\tau_{max}$ value will produce inclined cracks at an angle of 45 to the neutral axis, and this simulation results confirms the test results given in Figure 6.

In summary, crack development in the test is close to the result obtained by the simulation, and both belong to the shear failure mode.

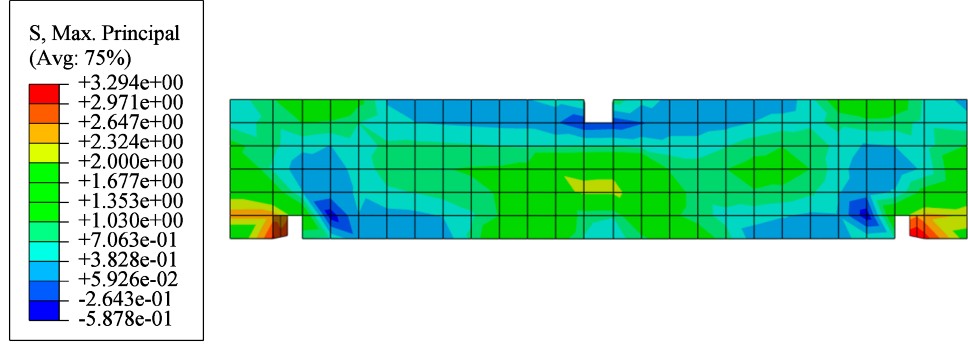

(**a**) Principal tensile stress contour plot of beam B-240

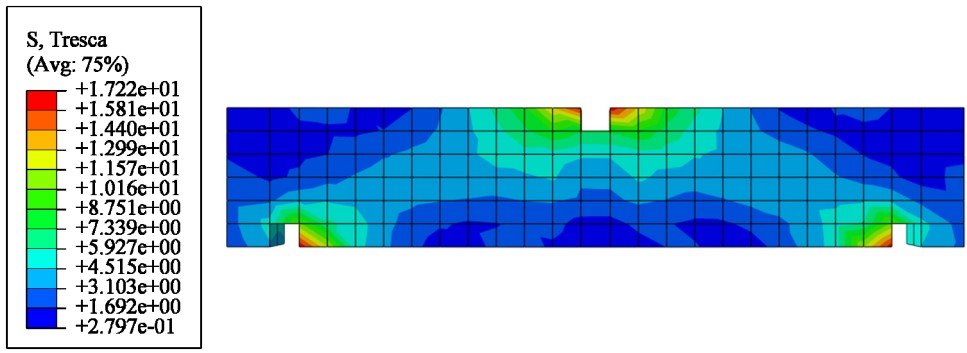

(**b**) Tresca stress contour plot of beam B-240

**Figure 7.** Characteristic stress contour plots.

### 4.2. Ultimate Load in Test and Simulation

The ultimate loads given by the simulation and experiment were very similar. Explicit and implicit algorithms were used to carry out the simulation analysis. The ultimate load $F_{u,s}^e$ was calculated by the explicit algorithm, and the ultimate load $F_{u,s}^i$ was calculated by the implicit algorithm, as shown in Table 10. In the table, the subscript t represents the test value, the subscript $s$ indicates the simulated value, the superscript $e$ represents the simulated value given by the explicit algorithm, and the superscript $i$ denotes the simulated value given by the implicit algorithm. In combination with the ultimate load $F_{u,t}$ given by the test, the relative error is $\varepsilon_{err}^i = \left|F_{u,s}^i - F_{u,t}\right|/F_{u,t}$ ($i = $ e, i). Obviously, for each beam, $\varepsilon_{err}^e$ is not only small but also much smaller than $\varepsilon_{err}^i$. This shows that the constitutive model used in this paper and the approximate solution algorithm of the selected nonlinear equations—the explicit algorithm—are reasonable. However, from the four sample points alone, there is clearly no qualitative relationship between $\varepsilon_{err}$ and the height of the beam $h$ (See Table 5).

**Table 10.** Numerical results for test and simulation.

| Specimen ID | $F_{u,t}$/kN | $F_{u,s}^e$/kN | $F_{u,s}^i$/kN | $\tau_t$/MPa | $\tau_s^e$/MPa | $\tau_s^i$ | $\varepsilon_{err}^e$ | $\varepsilon_{err}^i$ |
|---|---|---|---|---|---|---|---|---|
| B-120 | 44.00 | 47.06 | 30.96 | 1.67 | 1.78 | 1.17 | 6.95% | 29.64% |
| B-180 | 53.00 | 53.49 | 56.30 | 1.34 | 1.35 | 1.42 | 0.92% | 6.23% |
| B-240 | 67.00 | 70.88 | 29.80 | 1.27 | 1.34 | 0.56 | 5.79% | 55.52% |
| B-300 | 87.00 | 86.07 | 54.78 | 1.32 | 1.30 | 0.83 | 1.13% | 37.03% |

### 4.3. Law of Shear Strength and Its Value

Shear strength can be easily calculated based on the test and simulation results provided in the previous section. In concrete structure engineering, the shear strength of concrete beams is generally determined by the formula $\tau = V_u/bh_0$ [55], where $b$ is the

beam width and $h_0$ is the effective height of the cross-section of the beam—the two values for each specimen are shown in Table 5—and $V_u$ represents the ultimate shear bearing capacity of the cross-section. According to Figure 1, $V_u = F_u/2$, where $F_u$ is the ultimate load given by test or simulation (see Table 10 for specific values). According to the above, the calculated shear strength $\tau$ values are shown in Table 10.

The values of shear strength $\tau_t$ given by the test show an obvious size effect phenomenon. This phenomenon not only has regression statistical characteristics but can also be confirmed by the explicit simulation results, $\tau_{s,e}$. The $\tau_t$ (dependent variable), with $h_0$ as the independent variable under the constant shear span ratio ($\lambda = 2.45$) given by the experiment, is shown in Figure 8. Based on the set of discrete data points ($h_0$, $\tau_t$), the regression formula for describing the size effect is

$$\tau_t = 3.212 \times 10^9 \times h_0^{-4.856} + 1.279.$$

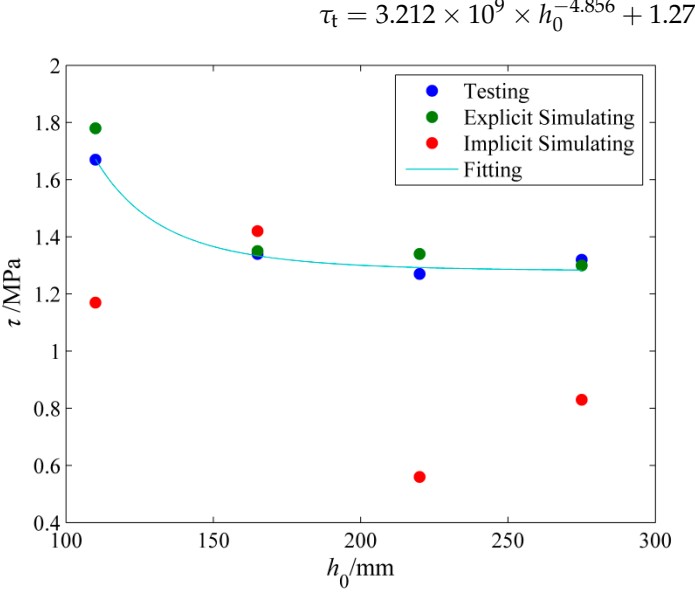

**Figure 8.** The test data of $\tau$ and its fitting curve.

The continuous curve of $h_0$ and $\tau_t$ obtained by the regression formula is shown in Figure 8; it is clearly seen that the fitting effect is qualitatively better. Either from the discrete data point set or from the fitting curve, it can be established that $\tau_t$ and $h_0$ are in a monotonous and nonlinear inverse proportional relationship: as $h_0$ increases, the deceleration rate of $\tau_t$ declines and shows a trend toward zero. Based on Table 10, the explicit simulation results are better than the implicit simulation results. The discrete data point set ($h_0$, $\tau_{s,e}$) given by the explicit simulation is shown in Figure 8. From the figure, it is easily derived that the law of explicit simulation data is the same as that of experimental discrete data, especially when $h_0$ is large. Herein, it is considered that this phenomenon has a certain scientific nature. As it is widely known, the accuracy of the approximate solution to the mathematical and physical problems (in this paper, the solid static problem) given by the structural finite element mainly depends on two factors: the ability to accurately describe the boundary conditions and the ability to accurately describe the general law of object behavior. Therefore, the explicit solution method provides a satisfactory solution for the finite element calculation of the shear strength of concrete beams described by the damage constitutive model.

## 5. Conclusions

In this paper, physical and simulation tests on the size effect of the shear strength of a group of recycled concrete beams without web reinforcement under the condition of a constant shear span ratio are carried out. Accordingly, the conclusions of this article are as follows:

1. From the test results, the shear strength of recycled concrete beams without web reinforcement has a size effect. In general, the shear strength is inversely proportional to the effective height of the section ($h_0$)—the smaller the $h_0$, the more obvious the size effect. In addition, the above-mentioned relationship can be better quantitatively described by the regression fitting formula.
2. Compared with the implicit finite element simulation results, the consistency between the explicit finite element simulation results and the experimental results is much higher. This situation not only shows that the damage constitutive model used in finite element modeling is reasonable, but it also shows that the explicit algorithm for solving this model is also reasonable.

This work will further promote the research of low-carbon-emission recycled concrete structures and the application of economic explicit finite element methods in static analysis of concrete structures.

**Author Contributions:** Conceptualization, W.W. and X.Z.; methodology, W.W. and X.Z.; software, X.Z.; formal analysis, W.W.; investigation, E.N.; resources, Y.-Q.G.; data curation, E.N.; writing—original draft preparation, X.Z.; writing—review and editing, W.W. and X.Z.; visualization, X.Z.; supervision, S-Q.C. and Q.-W.Y.; project administration, Y.-Q.G.; funding acquisition, Y.-Q.G. All authors have read and agreed to the published version of the manuscript.

**Funding:** This research was funded by Natural Science Foundation of Zhejiang Province (LY20E080012).

**Institutional Review Board Statement:** Not applicable.

**Informed Consent Statement:** Not applicable.

**Data Availability Statement:** Not applicable.

**Conflicts of Interest:** The authors declare no conflict of interest.

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
