# Peer review of "Size Effect of Shear Strength of Recycled Concrete Beam without Web Reinforcement: Testing and Explicit Finite Element Simulation"

_sustainability, doi:10.3390/su13084294_

Round 1
Reviewer 1 Report
- line 63, 'SRRC'
Please provide the full name.
- lines 87~88, 'implicit and ~ this system [36].
Still ambiguous. The difference between the implicit and explicit algorithms should be more clearly and concisely explained.
- lines 115~117
What is the main purpose of the paper, to present the size effect of SSRC, or to present the explicit method to evaluate the size effect?
If the first one is the main purpose, a more detailed investigation of the test results should be provided.
If the second one is the main purpose, a more clear explanation should be given about the implicit and explicit methods.
- Table 1 and the specimen names
In general, the beam depth or height represents the size of the beam. It is recommended to replace the spcimen names to include the beam depth or height, rather than the beam length.
- lines 169~179
In addition to the loading stages, please provide the loading speed, which could affect the shear strength of the beam.
- Figure 3
The meaning of the red concrete elements should be presented in the figure.
- Figure 4(b)
It looks not reasonable. Generally, concrete is considered as brittle, but the fracture energy due to the softening looks huge. In addition, in the case of shear-critical RC beams, the effect of the concrete softening behavior might be considerable. Please provide the background to support the tensile model in the figure.
- Figure 5(a)
Use the same format for the figures to present the crack patterns.
- Table 6
Check the notation for the superscript.
- line 368
Any comparison with the existing size effect models?
Reviewer 2 Report
The paper is of great interest, since it advances in the knowledge of the behavior of concrete with substitution of coarse aggregate for recycled concrete, when they are subjected to shear. The study is done experimentally and analytically, which makes it more interesting.
The paper is well structured and the conclusions are derived from the results obtained.
In order to improve the paper, some observations are made:
- The dosage used should be indicated at work, as well as the physical and chemical characteristics of the aggregates and cement.
- The method (number of specimens, type of specimens, etc ...) that have been used to obtain the compressive strength of the concrete, ?cp = 19.88MPa, and the modulus of elasticity ?c = 2.588 × 104MPa, should be explained.
- Why have you used the ratio ? = ?2 / ℎ0 = 2.45?
Reviewer 3 Report
The manuscript presents results from a numerical and experimental investigations on the use of recycled concrete with different sizes on shear strength beam without web reinforcement. The overall manuscript does not rise to the level of publication in the Sustainability Journal in the opinion of this reviewer. The manuscript was not very well prepared. There are a lot of holes. In fact, it reads like a report overall. The presentation is poor, and there lacks of in-depth analysis and discussions. The quality of the figures is low. Uncertainties are not provided for the reported test data. I cannot recommend it for publication before it is significantly improved. The following form the basis for this opinion.
- Abstract gives information on the main feature of the performed study, but some more details about the obtained results should be added. The abstract must be re-written. It is hard to follow; reading it makes an impression of randomly pasted sentences extracted from the text. It does not reflect properly the actual work done by the authors. I have a similar remark about conclusions.
-Authors must clarify necessity of this study. Aims and scope should be presented in the last part of introduction. (Research significance in presented, but it is not covered aims and scope of the research).
- How would the continuity of beam system in a full structure affect the results, compared to the tested simple spans? This can make significant difference in the shear and moment forces distributions. Please discuss and justify your experimental program/ FE analyses approach based on simple spans.
- The presented FE model was not validate with any experimental results. I suggest modeling the tested specimens and compare results with corresponding FE models results for validation.
- The conclusions presented are mainly a summary of the FE analyses and tests results. Design recommendations and feasibility need to be discussed further. The conclusion needs to be refined; it looks like a discussion.
- In order to better evaluate the advantage of using the recycled concrete in the beam system, comparing test results with code estimated capacities (shear and moment) is recommended. I would suggest comparing the ultimate bending and shear forces for the tested specimens with estimated code design values (e.g., ACI 318).
- The literature review merely lists the published work and does not present the major findings from each effort and how it ties to the presented research. The introduction does not identify the knowledge gap that the authors are trying to address with their research.
- The finite element analysis of data presented in the paper is a bit questionable and many of the paper's conclusions are not substantiated in the body.
- The objective statements are rather vague and lacks projected outcomes or how the paper will assist practitioners. Also, how does the recycled concrete help the shear beam behavior? Please consider to rewrite and clarify the motivation, objectives, and significance of this study.
- The finite element analysis of the data presented in the paper is a bit questionable. The use of large element size is enough for this reviewer to recommend decline of the manuscript until further justification for the element size and analyses are included. Despite significant efforts by authors, because of the fundamental flaw in the research method, findings of this study do not serve the purpose and objective of the research.
- The authors measured the two different deflections and the strains along with pressure transducers (see figure 2). However, there are no results presented and discussed in this study. The authors should provide the results of the deflections and strain gauges with the instrumentation plane along with the other measurements. The lack of discussion regarding the beam performance is enough for this reviewer to recommend decline of the manuscript until further results and discussions are included.
- Need to specify these parameters with specific physical meanings- most readers do not know ABAQUS details.
Round 2
Reviewer 1 Report
The review comments at the previous stage have not been reflected on the revised manuscript, in most cases. Only simple comments were relfected.
Although the study focused on the analytical approach with the 4 test results, most of the contents of the paper are not unique; the originality of the paper is very low. Even for the test results, only ultimate shear capacities were presented; usually, the load-deflection response should be compared when FE analysis is conducted, but nothing but the ultimate load was not included, although it was requested.
Moreover, the proposed equation considering the size effect should be validated with other test results regardless of using recycled material or not. If the model is significantly affected by the recycled material, the model is only available for the test specimen in the paper; not applicable for any other cases.
Reviewer 3 Report
The authors have made a considerable effort to revise the manuscript and the main comments about the scientific approach have been successfully addressed. The paper has been significantly improved however, there are several grammatical mistakes throughout the paper and English requires substantial editing to correct these mistakes and improve the quality of the written presentation using appropriate scientific English language. Also, there are many parts which need to be written concisely and revisions and refinements are required throughout the paper. This is important to ensure that the methodology and the findings of this work are successfully conveyed.
Also, Unfortunately, the part of numerical modelling has been improved only marginally, when this part needs to be fundamentally reworked, including the parts to the introduction. It is necessary to put the presented research into the context of the current state, to state the motivation and advantages of a used solution of numerical modelling, including inverse analysis. So far, the presentation of the results of numerical modelling is insufficient. The authors measured the two different deflections and the strains along with pressure transducers (see figure 2). However, there are no results presented and discussed in this study. The authors should provide at least one of the results of the deflections or strain gauges with the instrumentation plane along with the other measurements. The lack of discussion regarding the beam performance is enough for this reviewer to recommend a decline of the manuscript until further results and discussions are included. Also, the authors did not provide a clear answer about the use of large element size.